# Evening Meal Types and Family Meal Characteristics: Associations with Demographic Characteristics and Food Intake among Adolescents

**DOI:** 10.3390/nu12040886

**Published:** 2020-03-25

**Authors:** Francine Overcash, Cynthia Davey, Youjie Zhang, Marla Reicks

**Affiliations:** 1Department of Food Science and Nutrition, University of Minnesota, St. Paul, MN 55108, USA; mreicks@umn.edu; 2Clinical and Translational Science Institute, University of Minnesota, Minneapolis, MN 55414, USA; davey002@umn.edu; 3Department of Child and Adolescent Health and Social Medicine, Medical College of Soochow University, Suzhou 215123, Jiangsu, China; ujzhang@suda.edu.cn

**Keywords:** family meals, evening meal types, parents, adolescents, dietary intake

## Abstract

Evening meal types and family meal characteristics among adolescents may vary by demographic characteristics and weight status and can negatively impact dietary intake. We used cross-sectional Family, Life, Activity, Sun, Health, and Eating Study data from parent and adolescent dyads (12–17 years) to examine associations of adolescent evening meal types and family meal characteristics with adolescent and family demographic characteristics, weight status, and dietary intake. Multiple logistic regression models were used to evaluate associations of evening meal types and family meal characteristics with daily intake frequency of foods of interest, adolescent demographic characteristics, SES indicators, and weight status. All evening meal types that were not cooked from scratch showed associations with higher daily intake frequencies of less healthy food groups (fast food, convenience foods, sugar-sweetened beverages). Fast food eaten at home and heat-and-serve/box evening meals were associated with lower daily intake frequency of fruits and vegetables. Weight status, race/ethnicity, and age accounted for differences in associations with agreement regarding family meal characteristics. Agreement with often watching TV while eating, often eating alone and the importance of eating together were associated with daily intake frequency of some food groups. Evening meal types focused on convenience and negative family meal characteristics may adversely influence dietary intake among adolescents.

## 1. Introduction

Suboptimal diet quality of U.S. adolescents (ages 12–17) continues to be a serious public health issue. Compared to all other age groups, adolescents consume fewer fruits and vegetables [1,2], more sugar-sweetened beverages (SSBs) [3,4,5], and more fast foods [5,6]. Healthy Eating Index-2015 scores, which reflect alignment with the Dietary Guidelines for Americans [7], indicated that diet quality needed improvement among a U.S. nationally representative sample of adolescents [8]. Improvement was needed especially regarding low intakes of whole fruits, total vegetables, greens and beans, whole grains, seafood and plant proteins, and high intakes of refined grains, sodium, and added sugars and saturated fats.

Many factors influence food choices of adolescents during evening meals at home. Dietary intake at home is particularly dependent on what adults make available and expect adolescents to consume, especially among younger adolescents [9]. Convenience foods, or pre-prepared or partially prepared foods, are commonly available in home meals because they meet a need for quick and easy meal preparation when time is limited, yet counter a classically-defined home-cooked meal which generally connotes “made from scratch.” Convenience foods have been associated with lower diet quality based on higher saturated fat, sugar, and sodium content than foods that are unprocessed or require cooking [10,11] and are commonly consumed by U.S. adolescents [12]. A systematic review showed a positive association between ultra-processed food intake and body fat among adolescents [13]. Studies have shown that high cooking frequency or consumption of home-cooked meals was associated with better diet quality [14,15,16] and have characterized cooking frequency by sociodemographic characteristics [17]. However, these studies were conducted among adults who are typically the primary meal planners/preparers for the household. Limited studies have examined relationships between home cooking and diet quality of adolescents [18]. Other sources of foods consumed at home by adolescents that may negatively impact diet quality include fast food and foods delivered to homes. Foods prepared away from home have been associated with poorer diet quality among adolescents and adults [19,20]. Family meal characteristics, such as eating meals alone or watching TV during meals have also been attributed to lower adolescent diet quality [21,22,23,24].

As adolescents become young adults, they may perpetuate less healthful food acquisition or preparation habits common in their households. A first step to counter these negative habits is to better understand the relationship between evening meal types and adolescent diet quality based on the extent of food preparation. If evening meal types based on convenience foods and foods prepared away from home are associated with lower diet quality, educational programs could be implemented to decrease use of convenience foods and foods prepared away from home for evening meals at home. Programs that teach parents and adolescents food skills (meal planning and food purchasing based on nutritional considerations) and cooking skills could enable more health meals cooked from scratch and transference of skills from parents to adolescents.

The objectives of this study are to examine associations of adolescent evening meal types (cooked from scratch or a recipe, heat-and-serve/box meal, delivered, fast food) and family meal characteristics (perception of the importance of eating together, watching TV while eating dinner, eating alone) with adolescent demographic characteristics, weight status and dietary intake.

## 2. Materials and Methods 

This study was based on cross-sectional, web-based survey data from parent and adolescent (12–17 years) dyads sponsored by the U.S. National Cancer Institute (NCI) regarding type of evening meals eaten at home and family meal characteristics collected as part of the Family Life, Activity, Sun, Health, and Eating (FLASHE) Study in 2014 [25]. The purpose of the FLASHE Study was to better understand how lifestyle behaviors were associated with cancer risk. A non-probability sample was drawn from the Ipsos’s Consumer Opinion Panel to be similar to the U.S. population regarding sex, age, income, household composition, and region. Details on the design, sampling, recruitment, enrollment, and participant characteristics have been reported previously [26,27]. Adult panel members were screened for eligibility based on the following criteria: aged 18+, parent or legal guardian of an eligible adolescent, and living with the adolescent half of the time or more. One eligible adolescent was selected until quotas for three age groups were filled (12–13, 14–15, 16–17 years). About one-third of adolescent participants were recruited for each age group with one-half boys and one-half girls. Dyads were invited to complete diet and physical activity surveys. A public-use version of the data was available at the National Cancer Institute website, along with a codebook, data user’s guide, and a FLASHE methods summary report [28]. The FLASHE data are a quota sample from a consumer panel, therefore, survey weights were created as “analysis weights” to make the weighted sample more similar to the general population. Survey weights were constructed through raking procedures on key socio-demographic variables to reduce potential bias from population values [28]. The four raking variables used to construct weights for adolescents who completed the diet-related survey were age, gender, race/ethnicity, and Census region (Northeast, Midwest, South, West). According to the limitations described, statistical inference for variables other than these four raking variables are subject to selection bias, therefore, results of unadjusted associations cannot be generalized to the U.S. general adolescent population.

### 2.1. Participants

The sample size for analysis of evening meal types and family meal characteristics was 1657 adolescents, aged 12–17, with both diet-related survey data and demographic data from the parent diet-related/demographic survey. However, analysis sample size varied for each evening meal type/family meal characteristic outcome due to missing data from some teens. The University of Minnesota Institutional Review Board determined that analyses involving a de-identified, public-use version of the FLASHE Study data was exempt from review.

### 2.2. Demographic Characteristics and Weight Status

Adolescents reported their age, sex and race/ethnicity, height and weight. BMI percentiles were calculated according to Centers for Disease Control (CDC) guidelines [29] by FLASHE Study program investigators and used to classify adolescents into weight categories [28] as follows: underweight (BMI < 5^th^ percentile), normal weight (BMI 5^th^ - < 85^th^ percentile), overweight (BMI 85^th^ - < 95^th^ percentile) and obese (≥ 95^th^ percentile). Data from 66 underweight adolescents were used only in analyses that did not involve weight groups. For the current study, weight was dichotomized into 2 weight groups (normal weight vs. overweight/obese). The highest level of parent education completed and household income were obtained from parent report. FLASHE Study program investigators categorized highest level of parent education into 4 categories. Household income information was collected for 5 income categories and recoded by the FLASHE Study program investigators [28] into 2 categories ($0 to $99,999 and $100,000 or more).

### 2.3. Evening Meal Types

Adolescents were asked how many days in the past week (0–7) the evening meal eaten at home was comprised of each of the following types: (1) purchased from a fast food restaurant and eaten at home, (2) delivered to your home like pizza or Chinese food, (3) made from a heat-and-serve/box meal like Spaghetti-Os^®^, a microwave meal or frozen pizza and eaten at home, and (4) cooked from scratch or a recipe and eaten at home. Weekly frequency of each evening meal type was dichotomized by the number of evenings that maximized equal sample size distribution across comparison groups. Based on this distribution, the home evening meal types including purchased from a fast food restaurant, delivered, heat-and-serve/box meal, and cooked from scratch were dichotomized to “None” and “Any” in the past 7 days.

### 2.4. Daily Food Intake Frequency

Adolescents were asked to estimate their food intake frequencies “during the past 7 days” for each food and beverage item (27 total) with response options ranging from “I did not eat/drink this____” to “3 or more times per day.” Each item response was converted to a daily frequency: I did not eat/drink this____ = 0; 1–3 times in past 7 days = 0.29; 4–6 times in past 7 days = 0.71; 1 time per day = 1; 2 times per day = 2; 3 or more times per day = 3. Food and beverage items were then collapsed into categories based on a review of the literature considering cancer and obesity-related outcomes by the NCI FLASHE Study Program investigators [28]. For the current study, four food groups typical in adolescent diets were examined: junk food, convenience food, sugar-sweetened beverages (SSB) and fruit and vegetables. The category of *junk food* included candy/chocolate, cookies/cake, potato chips, fried potatoes, and frozen desserts. *Convenience foods* included fried potatoes, fried chicken, pizza, tacos, burgers, and heat-and-serve foods. *Sugar*-*sweetened beverages* included soda, energy drinks, sweetened fruit drinks and sports drinks. *Fruits and vegetables* included 100% fruit juice, fruit, green salad, other non-fried vegetables, cooked beans and other potatoes. Daily intake frequencies for items in these four food groups were summed to create food frequency scores that represent the aggregate daily intake for each food group by the FLASHE Study Program investigators and included in the publicly available data [28].

Adolescents with missing data for any item within a food group did not receive a score for that food group. As a result, daily intake food frequency scale scores were missing for 60–203 teens, depending on the scale. To deal with potential overestimation, daily intake frequencies for each food group were top-coded by the FLASHE Study Program investigators and included in the publicly available data [28]. A top-code or upper limit was set for daily intake frequency values based on consideration of possible overestimations of intake by adolescents. Values were considered overestimates if reported intakes corresponded to z-scores |≥ 3.29| (i.e., where 99.95% of scores would fall in a normal distribution). If the reported value corresponded with a z-score |≥ 3.29|, it was replaced by the value nearest to it without having a z-score |≥ 3.29| [28].

### 2.5. Family Meal Characteristics

For three statements, adolescents were asked to think about meal times with their family and rate their agreement about family meal characteristics with 5 response options from strongly disagree to strongly agree. The statements were: ‘In my family, it is important that we eat at least one meal together’, ‘In my family, we often watch TV while eating dinner’, and ‘In my family, I often eat alone.’ Each characteristic was dichotomized into agree (somewhat or strongly agree) vs. disagree/neutral (somewhat or strongly disagree/neither disagree nor agree) responses.

### 2.6. Data Analysis

Data were analyzed using SAS software (version 9.4, Cary, NC). Descriptive statistics (means and frequencies) were used to describe demographic characteristics, weight status, evening meal types, and dietary intake frequency of four adolescent food groups of interest: junk food, convenience food, SSBs, and fruits and vegetables. Logistic regression models, adjusted for adolescent age, sex, weight status, race, annual household income, parental education, and the four foods of interest (junk food, convenience food, SSBs, and fruits and vegetables), were used to examine associations of frequency (None vs. Any) of evening meal types (fast food at home, delivered meals, heat-and-serve/box meals, and meals prepared from scratch) and family meal characteristics (agree vs. disagree/neutral) with daily intake food frequencies of adolescent foods of interest, demographic characteristics, and weight groups. The top coded daily food frequency intake FLASHE survey variables were used in these analyses. A significance level of α = 0.05 was used to identify significant associations with evening meal type and family meal characteristics in adjusted regression models.

## 3. Results

### 3.1. Demographics, Mean Daily Intake Food Frequencies, Mean Number of Evening Meal Types, and Agreement with Family Meal Characteristics

One-third of the final dataset were 12–13 years old, about one-half were male, and almost one-third were overweight or obese (Table 1). The majority were Non-Hispanic white (64%) with 17% Non-Hispanic black and 10% Hispanic race/ethnicity. Most parents reported having an income < $100,000 (79%) and having some college or a college degree (82%). Adolescents reported higher mean daily intake frequencies per day of fruits and vegetables and junk food (2.8 and 1.9, respectively) than convenience foods and SSBs (1.5 and 1.3, respectively). They also reported a higher number of evening meals cooked from scratch per week (4.1) compared to evening meals purchased from a fast food restaurant and eaten at home (0.8), meals delivered to the home (0.4), and heat-and-serve/box meals (1.0). The majority of adolescents (75%) agreed that ‘In my family, it is important that we eat at least one meal together; half (49.5%) agreed that ‘we often watch TV during dinner’ and about one fifth (21%) agreed that ‘I often eat alone’ (Table 1).

### 3.2. Associations between Evening Meal Type and Adolescent Food Groups of Interest

Higher odds of adolescents reporting any fast food evening meals or evening meals delivered to the home in the past week were associated with increased daily intake frequencies of convenience foods and SSBs (Table 2). Higher odds of adolescents reporting any heat-and-serve/box evening meals in the past week were associated with increased daily intake frequency of junk foods and convenience foods. Lower odds of any fast food or any heat-and-serve/box evening meals in the past week were associated with decreased daily intake frequency of fruits and vegetables. Higher odds of any evening meals cooked from scratch in the past week were associated with increased daily intake frequency of fruits and vegetables and increased daily intake frequency of junk foods.

### 3.3. Associations between Evening Meal Type and SES Characteristics

Higher odds of adolescents reporting any fast food evening meals per week were marginally associated with each 1 year increase in adolescents’ age (Table 2). Hispanic adolescents were more likely to report any fast food evening meal or any meal delivered to the home in the past week compared to Non-Hispanic white adolescents (Table 2). Non-Hispanic black adolescents were more likely to report any fast food evening meal in the past compared to Non-Hispanic white adolescents (Table 2). Adolescents living in households with an annual income < $100,000 were more likely to report any evening meal in the past week that was made from a heat-and-serve/box compared to adolescents in households with annual income of $100,000 or more (Table 2).

### 3.4. Associations between Family Meal Characteristics and Adolescent Food Groups of Interest

Higher odds of adolescent agreement with the statement ‘It is important to eat at least one meal a day together” were associated with decreased daily intake frequency of convenience foods and increased daily intake frequency of fruits and vegetables (Table 3). Higher odds of adolescent agreement with “we often watch TV during dinner” were associated with increased daily intake frequency of convenience foods and SSBs, and decreased daily intake frequency of fruits and vegetables (Table 3). Higher odds of agreement with ‘I often eat alone’ were associated with increased daily intake frequency of convenience foods and decreased daily intake frequency of fruits and vegetables (Table 3).

### 3.5. Associations between Family Meal Characteristics and SES Characteristics

Odds of agreement with the statement, “It is important to eat at least one meal a day together” were lower for Non-Hispanic black adolescents (versus Non-Hispanic white) and for each 1 year increase in age (Table 3). Adolescents who identified as Hispanic, Non-Hispanic black, or Other (versus Non-Hispanic white), who were overweight/obese (vs. healthy weight), and who lived in households with an annual income < $100,000 (versus households with ≥ $100,000 annual income) had higher odds of agreement with the statement, “We often watch TV during dinner” (Table 3). Odds of agreement with the statement, “I often eat alone” were more likely for Non-Hispanic black adolescents (versus Non-Hispanic white), for overweight/obese adolescents (versus normal weight), and for each 1 year increase in age (Table 3).

## 4. Discussion

The current study examined the frequency of four typical evening meal types eaten at home and perceptions of three family meal characteristics and their associations with demographic characteristics, weight status and daily food intake frequencies of healthy and less healthy food groups among a nationwide sample of adolescents aged 12–17. A review of the relevant literature indicates this was the first description of the types of evening meals eaten at home among adolescents. In this nationwide sample, between two to three evening meals per week eaten at home were heat-and-serve/box meals, fast food meals, or food delivered to the home. Overall, the findings reflect the importance of the home environment regarding the healthfulness of adolescent meals and the need for improvement. The type of meal consumed by adolescents represents a behaviorally modifiable factor to inform obesity preventative interventions among this age group.

Findings from the current study indicated that diet quality, based on intake of four food/beverage groups typical in adolescent diets, was associated with evening meal types. For example, eating any fast food as an evening meal was associated with increased daily intake of less healthy food groups (e.g., convenience foods, SSBs), and decreased daily intake frequency of fruit and vegetables. These findings were likely because typical fast food meals include SSBs but not fruits and vegetables (exclusive of French fries) and because adolescents may be unlikely to ask for healthier alternatives to fast food meal components if available (e.g., no SSB, side of fruit/vegetable) due to taste and convenience. Adolescents with higher frequency of evening meals cooked from scratch had an increased likelihood of greater intake of fruits and vegetables per day because cooking meals from scratch may allow for more control of ingredients and the ability to modify recipes to include healthy ingredients such as fruits and vegetables. However, adolescents who reported higher frequencies of heat-and-serve/box evening meals per week were likely to have higher intakes of junk foods and convenience foods. Time constraints of families may necessitate an increased consumption of heat-and-serve/box meals, convenience foods, and fewer meals cooked from scratch. Therefore, interventions could promote improvements in dietary quality by offering ideas or strategies to make convenience foods more nutritionally sound choices. For example, strategies could include addition of vegetables that are easily incorporated (e.g., mixed-in or on the side) within these types of meals or how to read nutrition labels.

Diet quality of adolescents may be improved through interventions focused on food skills and cooking skills, but should be tailored to demographic characteristics based on the findings of the current study. For example, adolescents who identified as Hispanic or Non-Hispanic black may especially benefit from improvements in cooking skills given they were less likely than Non-Hispanic white adolescents to report any meal cooked from scratch in the past week. The general effectiveness of cooking skills interventions for increasing confidence in food preparation skills may contribute to more frequent meals cooked from scratch and improved dietary quality [30,31]. One cooking skills intervention among a racially and ethnically diverse population of parent-child pairs found an increase in cooking self-efficacy among young adolescents and preadolescents [32]. These strategies may be especially important for non-Hispanic black adolescents who were identified in the current study as less likely to endorse the importance of family meals and more likely to agree that they often watch TV during dinner or often eat alone compared to non-Hispanic whites. In addition, adolescents who practice screen time during meals or eat alone may benefit from innovative programming that utilizes technology to teach cooking skills. Food skills or the more recently coined term, *food literacy,* is a relatively new concept that focuses on a person’s ability to not only acquire food-related knowledge, but use knowledge to achieve better dietary outcomes [33]. A recent review of the use of technology in adolescent food literacy programs concluded technology could be a positive influence on dietary intake [34], as it caters to the preferences of this “digital native” age group.

Watching TV while eating family meals was associated with poorer adolescent diet quality and differed by sociodemographic characteristics. Adolescents completed the survey in 2014, prior to the overwhelming widespread use of smartphones. Therefore, watching TV may serve as a predecessor to screen time via personal devices of today. Watching any type of screen during meals may promote inadequate attention to what and how much is being eaten, which may lead to decreased dietary quality or increased intake of energy-dense foods. Screen-time during meals may also contribute to a less-formal and structured meal environment for adolescents, which is not conducive to positive family communication. The current study’s nationwide study population contributes evidence about screen viewing during family meals that appears to affect non-white, older adolescents, and those with overweight/obesity more negatively compared to Non-Hispanic white, normal weight, and younger adolescents. Family-focused interventions should consider tailoring strategies that limit screen time during meals by subgroups to improve adolescents’ diet quality.

This study had several limitations. The FLASHE project used cross-sectional data, which limited the ability to infer causation regarding effects of evening meal types and family meal characteristics on dietary outcomes. Analysis of income and dietary data were constrained to the categories provided by the FLASHE study investigators. For example, household income data were limited to a dichotomized variable (< $100,000 vs. $100,000+) and specific foods were assigned to food categories without the ability to modify (e.g., fried potatoes in junk food). In addition, the evening meal type questions could not differentiate whether eating alone or together. Responses were self-reported and therefore prone to bias. For example, regarding BMI, previous studies have shown under-reporting of weight and over-reporting of height for this age group [35,36].

## 5. Conclusions

Fast food meals, heat-and-serve/box meals, and delivered meals were commonly consumed as evening meals eaten at home by adolescents. The prevalence of consuming less healthy evening meal types and agreeing with adverse family meal characteristics differed by adolescents’ sociodemographic characteristics and weight status. Findings suggest that evening meal types and family meal characteristics are important factors associated with adolescents’ diet quality. Future intervention programs need to consider the importance of evening meal types and family meal characteristics and deploy relevant, feasible, and innovative strategies to improve adolescent diet quality. Potential effective strategies include improving the home food physical environment and adolescent cooking skills as well as limiting TV/screen viewing while eating dinner.

## Figures and Tables

**Table 1 nutrients-12-00886-t001:** Adolescent and parent/household characteristics, adolescent-reported mean daily food frequencies, and mean number of evening meal types in the past week.

Characteristics	Analysis Sample *n* = 1657
Adolescent	*n* (%)
Age	
12	219 (13.4)
13	326 (19.9)
14	276 (16.9)
15	288 (17.6)
16	326 (19.9)
17	202 (12.3)
Sex	
Male	810 (49.6)
Female	823 (50.4)
Weight group ^1^	
Normal weight	1085 (70.9)
Overweight/obese	446 (29.1)
Race/ethnicity ^1^	
Hispanic	160 (9.9)
Non-Hispanic black	272 (16.8)
Non-Hispanic white	1037 (64.0)
Other	152 (9.4)
Parent	
Household income	
< $100,000	1287 (79.3)
$100,000+	337 (20.8)
Parent education	
Less than high school/GED ^2^	21 (1.3)
High school/GED ^2^	272 (16.6)
Some college/no degree	577 (35.3)
4 year college degree or higher	767 (46.9)
Adolescent reported food type	Mean daily intake food frequency ^3^ (SD) ^4^
Junk food ^5^	1.9 (1.2)
Convenience foods ^6^	1.5 (1.0)
Sugar-sweetened beverages ^7^	1.3 (1.2)
Fruits and vegetables ^8^	2.8 (2.1)
Adolescent reported evening meal type	Mean number in the past week (SD) ^d^
Fast food ^9^	0.8 (1.1)
Delivered to home	0.4 (0.8)
Heat and serve/box meal ^10^	1.0 (1.4)
Cooked from scratch ^11^	4.1 (2.2)
Family meal characteristic	Adolescent agreement ^12^ *n* (%)
“It is important that we eat at least one meal together”	1241 (75.1)
“We often watch TV while eating dinner”	818 (49.5)
“I often eat alone”	343 (20.8)

^1^ Does not include 66 underweight adolescents ^2^ GED = General Equivalency Diploma ^3^ Top coded daily intake frequency in the past week ^4^ SD = Standard Deviation ^5^ Junk Food = candy/chocolate + cookies/cake + potato chips + fried potatoes + frozen desserts (*n* = 1569) ^6^ Convenience foods = fried potatoes + fried chicken + pizza + tacos + burgers + heat and serve (*n* = 1576) ^7^ Sugar-sweetened beverages = soda + energy drinks + sweetened fruit drinks + sports drinks (*n* = 1560) ^8^ Fruits and vegetables = 100% fruit juice + fruit + green salad + other non-fried vegetables + cooked beans + other potatoes (*n* = 1553) ^9^ Purchased from a fast food restaurant and eaten at home ^10^ Made from a heat and serve/box meal ^11^ Cooked from scratch or a recipe ^12^ Agreement = Somewhat agree/Strongly agree.

**Table 2 nutrients-12-00886-t002:** Odds ratios^1^ of evening meal types in the past week and daily intake frequencies of adolescent foods of interest and characteristics.

	Any Fast Food	Any Meals Delivered to Home	Any Heat-and-Serve/Box Meals	Any Cooked from Scratch Meals
Intake frequencies	Odds ratio (95% CI) (*p* value) ^2^	Odds ratio (95% CI) (*p* value) ^2^	Odds ratio (95% CI) (*p* value) ^2^	Odds ratio (95% CI) (*p* value) ^2^
Junk food: 1 time/day increase	1.06 (0.93, 1.20) (*p* = 0.408)	0.95 (0.83, 1.09) (*p* = 0.456)	1.15 (1.01, 1.30) (*p* = 0.040)	1.44 (1.12, 1.84) (*p* = 0.004)
Convenience foods: 1 time/day increase	2.22 (1.82, 2.71) (*p* < 0.001)	1.55 (1.31, 1.84) (*p* < 0.001)	2.41 (1.97, 2.95) (*p* < 0.001)	0.84 (0.64, 1.11) (*p* = 0.216)
SSBs: 1 time/day increase	1.18 (1.05, 1.33) (*p* = 0.007)	1.20 (1.07, 1.35) (*p* = 0.002)	1.07 (0.95, 1.21) (*p* = 0.243)	0.90 (0.75, 1.08) (*p* = 0.275)
Fruits and vegetables: 1 time/day increase	0.88 (0.82, 0.94) (*p* < 0.001)	1.00 (0.94, 1.07) (*p* = 0.984)	0.90 (0.85, 0.97) (*p* = 0.003)	1.30 (1.14, 1.47) (*p* < 0.001)
Demographics				
Age ^3^: 1 year increase	1.08 (1.00, 1.16) (*p* = 0.050)	0.98 (0.91, 1.07) (*p* = 0.664)	0.99 (0.92, 1.07) (*p* = 0.803)	1.05 (0.93, 1.18) (*p* = 0.459)
Sex: Male vs. female	1.00 (0.79, 1.27) (*p* = 0.997)	1.17 (0.91, 1.51) (*p* = 0.218)	0.96 (0.76, 1.22) (*p* = 0.751)	1.12 (0.76, 1.64) (*p* = 0.578)
Weight group: Overweight/obese vs. normal	0.98 (0.75, 1.27) (*p* = 0.861)	0.97 (0.73, 1.29) (*p* = 0.842)	1.17 (0.90, 1.53) (*p* = 0.236)	0.94 (0.62, 1.42) (*p* = 0.759)
Race: Non-Hispanic white (ref)				
Hispanic	1.53 (1.02, 2.30) (*p* = 0.039)	1.58 (1.05, 2.37) (*p* = 0.028)	0.89 (0.59, 1.34) (*p* = 0.584)	0.62 (0.35, 1.12) (*p* = 0.113)
Non-Hispanic black	1.80 (1.28, 2.53) (*p* = 0.001)	1.20 (0.85, 1.71) (*p* = 0.306)	0.78 (0.56, 1.10) (*p* = 0.156)	0.73 (0.44, 1.22) (*p* = 0.230)
Other	1.34 (0.88, 2.02) (*p* = 0.173)	1.16 (0.75, 0.18) (*p* = 0.502)	1.22 (0.80, 1.84) (*p* = 0.361)	0.80 (0.42, 1.50) (*p* = 0.481)
HH income: < $100K vs. $100K +	1.02 (0.75, 1.39) (*p* = 0.904)	0.78 (0.56, 1.07) (*p* = 0.122	1.55 (1.13, 2.11) (*p* = 0.006)	0.84 (0.50, 1.41) (*p* = 0.503)
Parent education: 4-year college degree or higher (ref)				
High school/GED	1.13 (0.79, 1.60) (*p* = 0.511)	1.00 (0.69, 1.46) (*p* = 0.986)	0.94 (0.66, 1.33) (*p* = 0.708)	1.09 (0.64, 1.86) (*p* = 0.760)
Less than high school/GED	1.58 (0.54, 4.64) (*p* = 0.409)	1.74 (0.63, 4.76) (*p* = 0.285)	0.61 (0.21, 1.81) (*p* = 0.371)	1.13 (0.24, 5.22) (*p* = 0.880)
Some college/no degree	1.02 (0.77, 1.35) (*p* = 0.884)	1.15 (0.85, 1.54) (*p* = 0.362)	1.03 (0.78, 1.37) (*p* = 0.818)	1.42 (0.90, 2.24) (*p* = 0.127)

^1^ Adjusted for all four food groups (junk, convenience, SSBs, fruits and vegetables), four demographic characteristics (sex, age, weight group, race), and two SES characteristics (annual HH income, parent education) ^2^ Significant at *p*-value < 0.05 ^3^ Odds of ‘Any’ meals per week for each 1 year increase in teen age.

**Table 3 nutrients-12-00886-t003:** Odds ratios^1^ of agreement with family meal characteristics and daily intake frequencies of adolescent foods of interest and characteristics.

	It’s Important to Eat at Least One Meal a Day Together	Often Watch TV during Dinner	I Often Eat Alone
Intake frequencies	Odds ratio (95% CI) (*p*-value)**^2^**	Odds ratio (95% CI) (*p*-value)**^2^**	Odds ratio (95% CI) (*p*-value)**^2^**
Junk food: 1 time/day increase	1.09 (0.95, 1.26) (*p* = 0.231)	0.99 (0.88, 1.13) (*p* = 0.913)	1.02 (0.89, 1.18) (*p* = 0.759)
Convenience foods: 1 time/day increase	0.74 (0.62, 0.89) (*p* = 0.001)	1.20 (1.02, 1.40) (*p* = 0.028)	1.37 (1.15, 1.65) (*p* < 0.001)
SSBs: 1 time/day increase	0.89 (0.79, 1.01) (*p* = 0.077)	1.29 (1.14, 1.45) (*p* < 0.001)	1.07 (0.94, 1.22) (*p* = 0.330)
Fruits and vegetables: 1 time/day increase	1.28 (1.18, 1.39) (*p* < 0.001)	0.89 (0.84, 0.95) (*p* < 0.001)	0.83 (0.77, 0.90) (*p* < 0.001)
Demographics			
Age ^3^: 1 year increase	0.89 (0.82, 0.97) (*p* = 0.005)	1.00 (0.93, 1.08) (*p* = 0.952)	1.10 (1.01, 1.20) (*p* = 0.036)
Sex: Male vs. female	1.16 (0.89, 1.52) (*p* = 0.278)	1.00 (0.80, 1.27) (*p* = 0.972)	0.78 (0.58, 1.03) (*p* = 0.080)
Weight group: Overweight/obese vs. normal	0.93 (0.70, 1.24) (*p* = 0.627)	1.41 (1.09, 1.83) (*p* = 0.008)	1.53 (1.14, 2.07) (*p* = 0.005)
Race: Non-Hispanic white (ref)			
Hispanic	1.05 (0.66, 1.69) (*p* = 0.832)	1.73 (1.17, 2.56) (*p* = 0.006)	0.87 (0.52, 1.45) (*p* = 0.580)
Non-Hispanic black	0.53 (0.37, 0.75) (*p* = 0.0003)	2.15 (1.54, 3.01) (*p* < 0.001)	1.82 (1.27, 2.61) (*p* = 0.001)
Other	0.71 (0.46, 1.14) (*p* = 0.138)	1.61 (1.08, 2.39) (*p* = 0.019)	1.28 (0.80, 2.07) (*p* = 0.307)
Household income: < $100K vs. $100K +	0.84 (0.59, 1.20) (*p* = 0.345)	1.40 (1.03, 1.89) (*p* = 0.030)	1.19 (0.81, 1.75) (*p* = 0.368)
Parent education: 4-year college degree or higher (ref)			
High school/GED	1.19 (0.80, 1.75) (*p* = 0.394)	1.21 (0.87, 1.70) (*p* = 0.262)	0.86 (0.57, 1.30) (*p* = 0.472)
Less than high school/GED	1.24 (0.42, 3.68) (*p* = 0.693)	1.30 (0.48, 3.49) (*p* = 0.610)	0.97 (0.33, 2.88) (*p* = 0.957)
Some college/no degree	1.19 (0.87, 1.62) (*p* = 0.273)	1.22 (0.93, 1.60) (*p* = 0.144)	0.92 (0.66, 1.28) (*p* = 0.616)

^1^ Adjusted for all four food groups (junk, convenience, SSBs, fruits and vegetables, four demographic characteristics (sex, age, weight group, race), and two SES characteristics (annual household income, parent education) ^2^ Significant at *p*-value < 0.05 ^3^ Odds of ‘Any’ meals per week for each 1 year increase in teen age.

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
