# Peer review of "Evening Meal Types and Family Meal Characteristics: Associations with Demographic Characteristics and Food Intake among Adolescents"

_nutrients, 2020, doi:10.3390/nu12040886_

Round 1

Reviewer 1 Report

The manuscript "Evening meal types and family meal characteristics: Associations with
Demographic characteristics and Food Intake among adolescents" is well written, correctly
analyzed and clear. However, as evidenced in the discussion, most of the results obtained confirm
previous studies. Therefore, what is the contribution to knowledge in the area? Maybe this is
partly because the data is from 2014 and for example TV is considered during meals and not
mobile phones, which are a big problem today at meals. Additionally, there is a significant
proportion of literature prior to 2010, which may be justifying the novelty of the study.
Therefore, authors must:
1. Update the bibliographic review and clearly identify the gap you would fill with your study.
2. Authors should make the effort to highlight which findings are original in their study and open
new lines of study, in the discussion section.
3. Propose how to carry out strategies to improve the home physical environment and adolescent
cooking skills. Regarding the latter, the authors must explain why they propose this. It would be
valid for teenagers who eat alone, but it is not the majority of the sample. Therefore, strategies to
improve adolescent feeding should be differentiated (and more detailed) according to the
problems presented by different groups of adolescents.

Author Response

Dear Nutrients’ Reviewer 1:

Thank you for the review of our manuscript, "Evening meal types and family meal characteristics: Associations with Demographic characteristics and Food Intake among adolescents.” Please see our point-by-point response in bold to each review comment below.

Reviewer 1:

The manuscript "Evening meal types and family meal characteristics: Associations with Demographic characteristics and Food Intake among adolescents" is well written, correctly analyzed and clear. However, as evidenced in the discussion, most of the results obtained confirm previous studies. Therefore, what is the contribution to knowledge in the area? Maybe this is partly because the data is from 2014 and for example TV is considered during meals and not mobile phones, which are a big problem today at meals. Additionally, there is a significant proportion of literature prior to 2010, which may be justifying the novelty of the study.

Therefore, authors must:

  1. Update the bibliographic review and clearly identify the gap you would fill with your study.

We reframed the introduction based on reviewers’ comments to make a stronger argument for the study objectives. In this process, we used more recent references to justify the need for this work. We are suggesting that a better understanding of the relationship between evening meal types and adolescent diet quality will enable more relevant educational programs for adolescents who will soon be young adults, given that previous work in this regard has mostly involved primary meal planners/preparers instead of adolescents.

  1. Authors should make the effort to highlight which findings are original in their study and open new lines of study, in the discussion section.

The entire discussion has been revised to address this comment. We have focused on our new findings and their implications rather than indicating how our findings confirm those from previous studies.

  1. Propose how to carry out strategies to improve the home physical environment and adolescent cooking skills. Regarding the latter, the authors must explain why they propose this. It would be valid for teenagers who eat alone, but it is not the majority of the sample. Therefore, strategies to improve adolescent feeding should be differentiated (and more detailed) according to the problems presented by different groups of adolescents.

The discussion has been revised to include strategies that would improve home environment and cooking skills of adolescents and corresponding justification for tailoring them to specific subgroups.

Reviewer 2 Report

Thank you for the invite to review for Nutrients and to the authors for their interesting read. The paper under consideration is relevant in scope to the journal. 

The paper may be suitable for publication, however I struggled to understand what the piece of research was bringing to the field that wasn't already known. We do have an existing understanding of the factors that influence evening meal type or family meal types (developed from quantitative and qualitative evidence). So the core novelty of the study is over-played here. I would encourage the reader to reframe, develop and make a stronger argument for the purpose of the paper building on what is already set up in the introduction going forward. This will help to sell why this paper matters and encourage more people to read it.

These issues may be an Editorial issue mind you as it is not for me to judge what is in scope for the journal and this issue may be less a problem here.

I also have some concerns over statistical decision making and study design. Some of this lies in a lack of clarity, others in decisions that are not justified or explained well. These include:

  • The demographic and weight status variables were all split into two groups with no clear rationale. Some may be fine (e.g. normal weight vs overweight/obesity), but others were not. Age was split into two groups due to 'equality' (line 120) but I do not understand this. Why not keep as a number and retain more information - the groups selected are not even in sample size (see Table 1) or number of years (2 v 4). Similar issues are present for education and income (particularly the latter where groups are very wide and include a heterogenous groups of individuals who are both very poor and middle class). These variables need reconsidering because they are all arbitrary and meaningless - yet they drive your results and having well thought out explanatory factors will help give a more meaningful set of results. This is probably why you find no consistent demographic patterns despite them being well established in the literature (e.g. SES differences)
  • Again the evening meal types were dichtomised to allow 'equal sample sizes' however that is not standard practice. Selection of variables should be based on theoretical implications (i.e. what they mean) not arbitrary statistical cut points. Reconsider these as will help maintain relevance of your findings.
  • Daily food intake frequency section is very confusing in the write up and needs improved clarity. Each item response was converted into a number, but there appears to be different item responses based on the description in line 143 which does not match the explanation in lines 141-2. Please revise.
  • I am not sure fried potatoes are junk food and certainly should not be included in multiple categories (I would keep in convenience over junk here)
  • It is not common practice to include fruit juices in measurements of fruits and vegetables consumption so I would remove them
  • Study limitations section (lines 323-327) is short and lacks depth - expand.

Minor:

Lines 70-73 - this is vague and repetitive so would suggest dropping

Line 188 - is underweight included in the normal weight category or dropped from analyis?

Line 203/Table 1 - There is a second 'N(%)' that can be removed lower down.

Tables 2 and 3 could be improved in presentation style - I found them dense and hard to follow. Revise.

Line 285 - I am less sure this is an educational issue (more a preference one). Given that you rubbish the statement in the next line, would leave out.

Author Response

Dear Nutrients’ Reviewer 2:

Thank you for the review of our manuscript, "Evening meal types and family meal characteristics: Associations with Demographic characteristics and Food Intake among adolescents.” Please see our point-by-point response in bold to each review comment below.

The paper may be suitable for publication, however I struggled to understand what the piece of research was bringing to the field that wasn't already known. We do have an existing understanding of the factors that influence evening meal type or family meal types (developed from quantitative and qualitative evidence). So the core novelty of the study is over-played here. I would encourage the reader to reframe, develop and make a stronger argument for the purpose of the paper building on what is already set up in the introduction going forward. This will help to sell why this paper matters and encourage more people to read it. These issues may be an Editorial issue mind you as it is not for me to judge what is in scope for the journal and this issue may be less a problem here.

We reframed the introduction based on reviewers’ comments to make a stronger argument for the novelty of the study objectives. Previous work regarding relationships between evening meal types, especially meals cooked at home, have mostly addressed primary meal planners/preparers rather than adolescents. We are suggesting that a better understanding of the relationship between evening meal types and adolescent diet quality will enable more relevant educational programs for adolescents who will soon be young adults. In addition, sociodemographic differences indicate that these programs should be tailored to meet needs of adolescent subgroups.

In addition to revising the introduction, we have revised the entire Discussion section to clarify how our findings have important implications for adolescent diet quality and to expand on what has been reported in previous literature.

I also have some concerns over statistical decision making and study design. Some of this lies in a lack of clarity, others in decisions that are not justified or explained well. These include:

  • The demographic and weight status variables were all split into two groups with no clear rationale. Some may be fine (e.g. normal weight vs overweight/obesity), but others were not. Age was split into two groups due to 'equality' (line 120) but I do not understand this. Why not keep as a number and retain more information - the groups selected are not even in sample size (see Table 1) or number of years (2 v 4). Similar issues are present for education and income (particularly the latter where groups are very wide and include a heterogenous groups of individuals who are both very poor and middle class). These variables need reconsidering because they are all arbitrary and meaningless - yet they drive your results and having well thought out explanatory factors will help give a more meaningful set of results. This is probably why you find no consistent demographic patterns despite them being well established in the literature (e.g. SES differences).

Age: We revised the age groups in Table 1 so that the ages are not collapsed and show the numbers and percentages by year of age. In addition, age was revised as a continuous variable in the regression models.

Education: Similarly, we revised the parent education variable to show the distributions for all 4 levels extablished by FLASHE study investigators. In addition, the regression models were revised to include the 4-level education variable.

Income: We were unable to revise the household income variable as these data were recoded by the NCI FLASHE investigators into two groups (<$100,000 and $100,000+). Only the recoded data are available to those who download and use the data, the original data were not available to us.

  • Again the evening meal types were dichotomized to allow 'equal sample sizes' however that is not standard practice. Selection of variables should be based on theoretical implications (i.e. what they mean) not arbitrary statistical cut points. Reconsider these as will help maintain relevance of your findings.

The 4 evening meal type outcome variables in our study were the original types provided by FLASHE study investigators. These variables are not completely independent variables because each asks about the number of evenings in the past 7 days the teen had each type of meal so the sum of the responses for these 4 variables is constrained at 7. Therefore, a higher number of evenings for one meal type necessarily results in a lower number of evenings for the other meal types. We have revised ALL evening meal type frequencies into None vs Any per week. We feel this categorization is appropriate because we are interested in odds ratios of characteristics by intake per week. 

  • Daily food intake frequency section is very confusing in the write up and needs improved clarity.

We followed the FLASHE Data User’s Guide to describe how the daily food intake frequency was calculated with revisions to clarify.

Each item response was converted into a number, but there appears to be different item responses based on the description in line 143 which does not match the explanation in lines 141-2. Please revise.

We revised the description as follows:  Adolescents were asked to estimate their food intake frequencies “during the past 7 days” for each food and beverage item (27 total) with response options ranging from “I did not eat/drink ____” to “3 or more times per day.” Each item response was converted to a daily frequency: I did not eat/drink ____ = 0; 1-3 times in past 7 days = 0.29; 4-6 times in past 7 days = 0.71; 1 time per day = 1; 2 times per day = 2; 3 or more times per day = 3.

  • I am not sure fried potatoes are junk food and certainly should not be included in multiple categories (I would keep in convenience over junk here)

We used the same food and beverage categories designated by the NCI FLASHE investigators to categorize food and beverage items. It was not possible to change food or beverage categorizations from one group to another. The FLASHE User’s Guide describes the rationale for their categorization:

Estimating daily frequency has the advantage of being relatively easy to calculate and allows a researcher to combine foods into food groups for which a common unit of measure (other than daily frequency) does not exist. This is useful for estimating intake from more heterogeneous food groups such as “junk food.” For FLASHE, food groups such as several sub-categories of “detrimental” foods, were quantified using this approach because they do not share a standard common unit for quantification, other than daily intake frequency, and they are similar in characteristics such as method of consumption (e.g., beverages) and/or primary ingredients (e.g., sugary foods). The FLASHE survey defines junk food as foods that are high in calories and usually have added sugars and fat and include candy, cookies, potato chips, French fries, etc. Sugar sweetened beverages (SSB) (i.e., sugary drinks) are defined as regular soda, sports drinks, fruit drinks, sweetened teas and other drinks with added sugar. Additional groups were not defined in the survey, but were formed by grouping similar FLASHE items. This approach does not integrate metrics such as portion size, so it cannot be used to estimate precise intake of foods or nutrients. Caution should be exercised in interpretation, as these values are not rooted in any comparative values. For example, there are no specific recommendations for frequency of intake of junk food, and junk food is not universally defined.

Food and beverage items were then collapsed into categories based on a review of the literature considering cancer and obesity-related outcomes by the NCI FLASHE Study Program investigators.

  • It is not common practice to include fruit juices in measurements of fruits and vegetables consumption so I would remove them.

Similar to fried potatoes in junk food, 100% fruit juice was included in the FLASHE definition of Fruits and Vegetables and therefore cannot be removed from the fruits and vegetables variable. This limitation has been added to the Limitations section.

  • Study limitations section (lines 323-327) is short and lacks depth - expand.

We have expanded the limitations section per your request. We focused additional information on the inability to modify how foods and beverages were categorized by NCI FLASHE Study Program investigators.

Minor:

Lines 70-73 - this is vague and repetitive so would suggest dropping.

These lines have been deleted.

Line 188 - is underweight included in the normal weight category or dropped from analysis?

In the methods section 2.2 Demographic characteristics and weight status, we indicated: Data from 66 underweight adolescents were not used in analyses involving weight groups. Therefore, underweight was not included in the normal weight category and underweight was only dropped from analysis that include that weight groups.

Line 203/Table 1 - There is a second 'N(%)' that can be removed lower down.

We removed the second N (%) for Parents.

Tables 2 and 3 could be improved in presentation style - I found them dense and hard to follow. Revise.

We have revised and reformatted tables to make them easier to follow.

Line 285 - I am less sure this is an educational issue (more a preference one). Given that you rubbish the statement in the next line, would leave out.

We removed the following statement: This may also be suggestive of a lack of basic nutritional knowledge for this age group.

Round 2

Reviewer 1 Report

The authors improved the manuscript, so I suggest to accept it.

Reviewer 2 Report

All changes made - I thank the authors for their hard work. No further suggestions here.